# Numerical Study and Structural Optimization of Water-Wall Temperature-Measurement Device for Ultra-Supercritical Boiler

**DOI:** 10.3390/s24186038

**Published:** 2024-09-18

**Authors:** Zifu Shi, Pei Li, Yonggang Zhou, Song Ni

**Affiliations:** Institute of Thermal Power Engineering, Zhejiang University, Hangzhou 310027, China; 0914166@zju.edu.cn (Z.S.); trooper@zju.edu.cn (Y.Z.); 22327111@zju.edu.cn (S.N.)

**Keywords:** water-wall, temperature measurement, structural optimization, CFD

## Abstract

The temperature of the water wall in the furnace chamber is extremely important for the daily operation of a boiler. Considering the high temperature and dusty environment in the furnace, a temperature measurement device mainly composed of four parts (armored temperature sensor, in-furnace heat-collecting block, out-furnace fixing base, and protective cannula) was designed in this study, which could be used to directly obtain the temperature of the in-furnace water-wall. Numerical simulations of temperature measurement devices with different heat-collecting block structures were carried out using the computer fluid dynamics method. After comparing the measurement accuracy and considering the practical application scenarios, the optimized heat-collecting block structure with a specific expansion gap (0.5 mm wide and 4 mm deep) was selected for practical application. Such a temperature measurement device was then applied to a 1000 MW ultra-supercritical coal-fired boiler in China, and the tested in-furnace water-wall temperature data were in good agreement with relevant research. Compared with the conventional temperature measurement device arranged outside the furnace, the in-furnace water-wall temperature-measurement device adopted in this study has a more sensitive response characteristic and can directly reflect the temperature of the water wall inside the furnace. In addition, it can also reflect the local slag formation state of the water wall and has a long service life.

## 1. Introduction

In 2023, China’s coal-fired power-generation capacity reached 5.35 × 10^12^ kW·h, accounting for 63% of the country’s overall power generation [1]. On a global scale, coal-fired power plants currently operational in China account for 50% of the world’s total [2,3]. Given this dominance of coal-fired power generation, there is no denying that the reliable and safe operation of boilers—critical components of coal power units—plays a decisive role in ensuring the stability of China’s and the world’s power market. Globally, boiler accidents are a critical safety concern, and related incidents have been reported in countries such as China, Malaysia, and South Africa, resulting in significant financial losses and casualties [4,5,6]. Notably, a primary cause of these accidents is damage to the heating surface of the water wall [7]. The increasing involvement of coal-fired units in peaking operations results in significant changes in the flow dynamics, thermal physical characteristics, and thermal stress distributions of working media within the water walls of coal-fired boilers. Furthermore, frequent load variations lead to dramatic changes in the flow and temperature fields of flue gases within these boilers. These changes, in turn, lead to alternating fluctuations in the temperature and stress distributions on the furnace water-wall, often resulting in transverse cracks on the tube wall surface, eventual tube wall rupture, and potential tube explosion [8]. Generally, the temperature of the fireside water wall is a direct indicator of whether the water-wall temperature lies within the permissible limits established for the tube. And this temperature directly impacts the yield strength and creep rate of the tube. Thus, to prevent above-mentioned accidents and consequently ensure the safe operation of boilers, monitoring the temperatures of the water-wall tubes is critical.

Currently, several methods have been adopted for monitoring the temperature of the water wall. For instance, Wen and Fu et al. [9,10] investigated the temperature, heat flux, strain, and thermal stress distributions of the water wall of a coal-fired boiler at the laboratory scale. Their findings highlighted that the confinement location affected the stress and strain distributions of the water wall; however, their observations may not be entirely accurate owing to substantial discrepancies between laboratory conditions and real-world industrial processes. Furthermore, Wang et al. [11] assessed the safety of heat transfer processes in the low-mass-flux vertical water wall of a high-load supercritical coal-fired boiler in the laboratory. Their results revealed good heat transfer characteristics of the experimental water-wall tube under conditions of low mass and standard heat fluxes. However, as the test pressure approached critical levels, both the metal temperature and pressure drop across the water-wall tube increased rapidly and abruptly.

The finite element analysis approach is widely considered an effective method for assessing the thermal characteristics of boiler water walls [12,13,14,15]. In research focused on water-wall temperature monitoring, Zhang et al. [16] examined the thermal stress and deformation distributions within the steam section of the water wall of a 660 MW boiler under two typical working conditions using numerical simulations. Their analysis revealed the temperature distributions surrounding the water-wall tube. Lv et al. [17] introduced a multiple-model adaptive inverse scheme based on boundary condition transfer, designed to estimate time-varying thermal boundary conditions and reconstruct temperature fields on the firesides of membrane water walls. The effectiveness of this approach was validated using numerical simulations. Additionally, Pang et al. [18] employed computational fluid dynamics (CFD) to analyze the temperature differences and thermal stress acting on the water wall of a boiler. Their results highlighted significant water-wall temperature differences during boiler startup, ranging from 80 to 150 °C. Furthermore, through experimental research and simulations, Zhou et al. [19] employed a new PRT model to assess heat transfer safety in the water wall of a 660 MW ultra-supercritical circulating-fluidized-bed boiler. Their results revealed that using a smooth tube design for the wate wall could prevent film boiling and overheating, ensuring safe heat transfer. Zima et al. [20] and Dong et al. [21] developed mathematical models to examine the temperature distributions of the water walls of supercritical boilers. Their findings revealed that the heat flux within the furnace critically influenced the thermal performance of the water wall. Furthermore, they observed that the heat flux distribution of the water wall remained nearly constant and demonstrated a positive correlation with the water-wall height. Here, the maximum temperature of the wall was found to be below 803 K, which lies within the allowable temperature limit for the material under ideal conditions.

The aforementioned studies examining water-wall temperatures either rely on laboratory tests or theoretical calculations, which present inherent limitations. The calculation procedures in the indirect measurements can bring extra measurement uncertainty, which impedes the utilization and development of the indirect measurements. Direct measurements in this context are particularly difficult owing to the high temperatures and dusty environments of furnaces, resulting in a lack of relevant research. Typically, studies examining the water-wall temperatures of power plants rely on the readings of temperature sensors installed on the outer surfaces of the water walls. However, these temperature readings recorded outside the furnace do not immediately and accurately reflect the true temperatures of the furnace’s inner wall. Consequently, it is necessary to develop a new method that can directly measure the inner wall temperature of water wall and protect the temperature sensor from abrasion in coal-fired boilers.

In this study, a wall temperature-measurement device composed of four parts (armored temperature sensor, in-furnace heat-collecting block, out-furnace fixing base, and protective cannula) was specially designed, which could be used to directly obtain the temperature of the in-furnace water wall. A numerical simulation approach was employed to optimize the structure of its heat-collecting block. Then, the water-wall temperature-measurement device with optimal structure was applied to a 1000 MW class boiler in China, and its reliability was validated. Overall, by offering direct measurements of the inner wall temperatures of the water wall, the designed device can assist operators in accurately regulating the operating conditions of boilers, thereby increasing the service lives of heating surfaces and boosting the safety and economic efficiency of boilers.

## 2. Device Description

### 2.1. Structure

As illustrated in Figure 1, the temperature-measurement device designed in this study to accurately capture the inner water-wall temperature of a boiler primarily comprises four components: an armored temperature sensor, an in-furnace heat-collecting block, an out-furnace fixing base, and a protective cannula. Notably, similar structures have been adopted in another study to monitor the slagging state of water walls; however, this application possesses less stringent accuracy requirements for temperature measurements [22].

During installation, the in-furnace heat-collecting block and out-furnace fixing base are welded to the fin between two adjacent water-wall tubes. The measuring end of the armored temperature sensor is first inserted through holes drilled into the fixing base and fins and subsequently passed through a preset hole on the heat-collecting block. The protective cannula is connected to the fixing base using threads, ensuring the secure placement and protection of the temperature sensor in the complex operating environment of the boiler.

Notably, the in-furnace heat-collecting block is designed to safeguard the temperature sensor from direct furnace radiation as well as coal-ash-particle-induced erosion and wear. The in-furnace heat-collecting block, out-furnace fixing base, and membrane water-wall fins are all made from the same material to prevent issues related to variations in expansion coefficients. The heat-collecting block and fixing base are exclusively welded to the fins to avoid the potential safety risks associated with their integration into the high-pressurized water-wall tubes. The inner arc of the heat-collecting block precisely matches the surface curvatures of the water-wall tubes. Expansion gaps are strategically designed to ensure continuous and complete contact between the heat-collecting block and adjacent water-wall tubes during device installation and boiler operation. This design minimizes thermal resistance and improves measurement accuracy. Remarkably, the aforementioned temperature measurement device, featuring a simple structure, easy installation, and low cost, offers direct measurements of the water wall temperature on the fireside.

### 2.2. On-Site Installation

The on-site installation of the above-mentioned temperature measurement device on the water wall is depicted in Figure 2. To ensure the safety of the water-wall tubes, heat-collecting blocks and fixing bases were spot-welded with the fins.

## 3. Calculation Method

### 3.1. Geometry

To investigate the impacts of different structural parameters of the heat-collecting block on the water-wall temperature-measurement device, a three-dimensional calculation model was established, as depicted in Figure 3. Notably, this model is based on a 1000 MW ultra-supercritical coal-fired boiler in China, and the key dimensions of its membrane water walls are summarized in Table 1.

The wall temperature measurement device was incorporated into the calculation domain, which was designed to be sufficiently large to satisfy the requirements of numerical simulations. In this model, the flows of high-temperature flue gas within the furnace and the working medium within the water wall were directed from bottom to up. By comparing the calculated temperature at the measuring end of the heat-collecting block with that of the water-wall tubes, structures with high measurement accuracy were identified. Practical application scenarios were then considered to identify the most suitable structure.

### 3.2. Meshing

ANSYS meshing was employed to divide the simulation domain into tetrahedral meshes, locally encrypt the simulation domain, and establish boundary layers on the walls of the fluid domain. To eliminate the influence of mesh count on the obtained results, a mesh independence verification analysis was conducted. As illustrated in Figure 4 and Figure 5, under the same boundary conditions, varying mesh counts—1,653,818; 3,035,154; and 4,764,331—yield consistent temperatures at both the measuring end of the temperature sensor and the top of the fireside of each water-wall tube. Thus, the mesh count does not significantly affect the calculation outcomes. For computational convenience, a mesh count of approximately 3 × 10^6^ was selected for the simulation analysis. The final mesh configuration is depicted in Figure 6, with a mesh quality exceeding 0.3.

### 3.3. Mathematical Model

The ANSYS Fluent was used for numerical simulation. During simulation, the structural thermal stress model was opened, the Realizable *k*-*ε* turbulence model was adopted for the viscous model, and the standard wall function was adopted. Since the flue gas components in the simulated furnace contain three atomic gases, the influence of radiation on heat transfer must be considered. By using the radiation model (DO) model for radiation, using the SIMPLE method for coupling equations algorithm to solve the pressure—speed coupling equation. The discrete form of the governing equation is the second-order upwind scheme. The numerical simulation mathematical description of the method for measuring the in-furnace water-wall-temperature is as follows:

#### 3.3.1. Basic Equation

The continuity equation, the momentum equation, and the energy equation are regarded as the foundation formulas in the field of computational fluid mechanics.

Mass conservation equation:(1)$$\frac{\partial \rho }{\partial t}+\frac{\partial \rho u_{i}}{\partial x_{i}}=0$$

Momentum conservation equation:(2)$$\frac{\partial \left(\rho u_{i}\right)}{\partial t}+\frac{\partial }{\partial x_{j}}\left(\rho u_{i}u_{j}\right)=\frac{\partial \tau_{ij}}{\partial x_{j}}-\frac{\partial p}{\partial x_{i}}+\rho g_{i}+F_{i}$$

Energy conservation equation:(3)$$\frac{\partial \left(\rho c_{p}T\right)}{\partial T}+\frac{\partial }{\partial x_{j}}\left(\rho c_{p}u_{i}T\right)=\frac{\partial }{\partial x_{i}}\left(\lambda \frac{\partial T}{\partial x_{j}}-\rho c_{p}u_{j}^{\prime }T^{\prime }\right)+S$$
where p represents static pressure, $\tau_{ij}$ represents viscous stress, ρ represents density, $F_{i}$ represents external stress in direction $i\rho g_{i}$ represents gravity volumetric force in the direction *i*, $u_{i}$ represents speed in direction *i*, and $u_{j}^{\prime }$ represents pulse velocity in direction *i*.

#### 3.3.2. Gas-Phase Turbulent Flow Model

The gas flow in the furnace is a three-dimensional turbulent reaction flow, and its average flow can be regarded as a steady flow, so it can be described by the usual conservation equation. For the more mature turbulence model used in industry, the two-equation *k*-*ε* model can be used. Any instantaneous physical quantity in the N-S equations is expressed in the form of the sum of the average quantity and the pulsation quantity, and the time average equations of turbulent motion can be obtained by performing the time average operation on the whole equations.
(4)$$\frac{\partial \left(\rho k\right)}{\partial t}+\frac{\partial \left(\rho ku_{i}\right)}{\partial x_{i}}=\frac{\partial }{\partial x_{j}}\left[\left(\mu +\frac{\mu_{t}}{\sigma_{k}}\right)\frac{\partial k}{\partial x_{j}}\right]+G_{k}+G_{b}-\rho \varepsilon -Y_{M}+S_{k}$$
(5)$$\frac{\partial \left(\rho \varepsilon \right)}{\partial t}+\frac{\partial \left(\rho \varepsilon u_{i}\right)}{\partial x_{i}}=\frac{\partial }{\partial x_{j}}\left[\left(\mu +\frac{\mu_{t}}{\sigma_{\varepsilon }}\right)\frac{\partial k}{\partial x_{j}}\right]+C_{1\varepsilon }\frac{\varepsilon }{k}\left(G_{k}+C_{3\varepsilon }G_{b}\right)-C_{2\varepsilon }\rho S_{\varepsilon }$$
where $G_{k}$ represents the generation term of turbulent kinetic energy caused by the average velocity gradient, $G_{b}$ represents the amount of turbulent kinetic energy generated by buoyancy, $Y_{M}$ represents the influence of compressible turbulent pulsating expansion on the total loss rate, $C_{1\varepsilon }$, $C_{2\varepsilon }$, $\sigma_{k}$, $\sigma_{\varepsilon }$ represents general empirical constant, and μ represents dynamic viscosity.

#### 3.3.3. Radiation Heat Transfer Model

The Discrete Transfer Method proposed by Lockwood and Shah et al. is used to calculate radiative heat transfer. When considering the temperature T of the medium in the grid and the absorption, emission and isotropic scattering of the radiant energy beam by gas, the intensity change in the radiant energy beam passing through a grid is as follows:(6)$$\frac{dI}{dt}=-K_{a}I+\frac{\sigma }{\pi }K_{a}T^{4}$$
where $K_{a}$ represents the absorption coefficient of gas.
(7)$$K_{a}=0.28\exp\left(-T/1135\right)$$

#### 3.3.4. Von Mises Model

Von Mises stress (also called equivalent stress), which adhered to the fourth strength theory of material mechanics, is regarded as one of the yield criteria. Stress isoline is used in the Von Mises model to express the stress distribution so that the unsafe area in the model can be determined precisely. The Von Mises model is commonly utilized when the general material is deformed through the action of external force or destroyed by the flow movement. The relationship between the Von Mises stress and the principal stresses is as below:(8)$$\sigma_{e}=\sqrt[{2}]{0.5\left[{\left(\sigma_{1}-\sigma_{2}\right)}^{2}+{\left(\sigma_{2}-\sigma_{3}\right)}^{2}+{\left(\sigma_{1}-\sigma_{3}\right)}^{2}\right]}$$
(9)$$\varepsilon_{e}=\frac{1}{1+\nu^{\prime }}\sqrt[{2}]{0.5\left[{\left(\sigma_{1}-\sigma_{2}\right)}^{2}+{\left(\sigma_{2}-\sigma_{3}\right)}^{2}+{\left(\sigma_{1}-\sigma_{3}\right)}^{2}\right]}$$
where σ represents stress, ε represents strain, and $\nu^{\prime }$ represents elfective Poisson’s ratio, which is defined as material Poisson’s ratio for elastic and thermal strains estimated at the reference temperature of the body, or 0.5 for plastic strains.

### 3.4. Material Properties

The water wall, the in-furnace heat-collecting block, and the out-furnace fixing base are all made of 15CrMo, and the mass percentage (%) of its constituent elements is shown in Table 2. The physical properties of 15CrMo, such as yield strength, allowable stress, and elastic modulus at 20 to 600 °C, are shown in Table 3.

### 3.5. Boundary and Working Conditions

During the simulation, boundary conditions were set at the flue gas inlet, flue gas outlet, working medium inlet, and working medium outlet. In this context, the inlet and outlet conditions were defined as velocity inlet and pressure outlet boundary conditions, respectively, and the wall was modeled as a non-slip surface. To simulate a suspended boiler water wall, the upper plane of the water wall was constrained by thermal expansion, while the remaining parts were allowed to deform freely. Furthermore, the boundary conditions for steam and flue gas were defined based on the parameters of the selected 1000 MW ultra-supercritical boiler.

As shown in Figure 7 and Figure 8, to examine the influence of different structural parameters on temperature measurements, the thickness of the heat-collecting block and the location, width (c), and depth (c) of the expansion gap were varied while maintaining the parameters of the flue gas and working medium constant. The specific working condition settings are outlined in Table 4.

## 4. Results and Discussion

### 4.1. Flow Simulation Results

Considering Case 1 as an example, the temperature and velocity distributions within the calculation domain are depicted in Figure 9. In this system, heat is transferred from the flue gas to the working medium through the water-wall tubes. Accordingly, as the flue gas traverses the system, its temperature decreases from 1318.0 °C at the inlet to 1291.4 °C at the outlet. Conversely, the temperature of the working medium increases slightly from 377 °C at the inlet to 377.3 °C at the outlet. Notably, owing to the small height of the water wall, changes in the temperatures of both the working medium and flue gas are minimal. In the heat-collecting block, flue gas flows from bottom to top. Specifically, intense heat exchange interactions occur between the lower section of the heat-collecting block and flue gas. Meanwhile, in the upper section of the heat-collecting block, a reflux zone is created (Appendix A) characterized by low flue gas temperatures and weaker heat exchange interactions. These differences result in higher temperatures in the lower section of the heat-collecting block compared to those in the upper section. Temperature gradient distributions reveal that the temperature decreases gradually from the flue gas to the working medium (Appendix A). Furthermore, the greater the proximity of the heat-collecting block to the fireside of the boiler, the greater its temperature. Figure 10 depicts the temperature distribution on the surface of the water wall, which aligns with the trends reported by previous related studies [25]. Overall, our findings align with the requirements of stable industrial operations; consequently, our numerical modeling approach can be reasonably applied to analyze the thermal stress and temperature distributions of both the water wall and heat-collecting block.

### 4.2. Thermal Expansion in the X Direction (αL)

Because the water wall is suspended, the vertical thermal expansions of both the in-furnace heat-collecting block and the water wall are similar. However, the primary factor affecting temperature measurements is the expansion along the X direction (Figure 11). Under varying operating conditions, this expansion can introduce unnecessary contact thermal resistance, impacting the heat exchange between the heat-collecting block and water-wall tubes. Hence, our subsequent analysis of temperature measurement focuses solely on the thermal expansion along the X direction. Given the symmetry of the heat-collecting block, any one side of this block (defined as Surface 1) is selected for analysis. The location of Surface 1 on the heat-collecting block is illustrated in Figure 11, with 6 lines marked on it.

According to the solid thermal expansion theory [26], the linear expansion of solids is calculated as follows:(10)dl=α·l·dT
where α denotes the linear thermal expansion coefficient, l represents the linear size of the sample, and T denotes the temperature.

This equation indicates that the linear thermal expansion of a solid is directly proportional to the coefficient of expansion, linear dimension, and temperature change of the sample. Figure 12 illustrates the thermal expansion of Surface 1 along the X direction. Evidently, the greater the proximity of Surface 1 to the flame, the greater its thermal expansion, which is attributed to its enhanced heat exchange with the flame. Because the flue gas traverses the heat-collecting block from bottom to top, its heat exchange interactions with the lower section of the heat-collecting block are more intense compared to those with the upper section, resulting in higher temperatures and greater thermal expansion in this region. In particular, the maximum thermal expansion occurs at the lower right of Surface 1. Furthermore, the temperature gradient of Surface 1 decreases from top to bottom, consistent with the theory of solid thermal expansion. Under the same thickness of the heat-collecting block, Case 3 exhibits a smaller overall thermal expansion, whereas Case 6 exhibits a larger overall thermal expansion. Furthermore, Cases 8 and 9, featuring larger contact areas with the flue gas, exhibit enhanced heat transfer and the highest thermal expansion. The thermal expansion of Case 2 surpasses that of Case 1, whereas the thermal expansion of Case 3 is lower than those of Cases 1 and 2. Among Cases 2, 4, and 5, the maximum expansion of Case 5, exhibiting the widest expansion gap, is slightly lower than those of Cases 2 and 4. Meanwhile, in Cases 2, 6, and 7, featuring varying expansion gap depths, the maximum expansion of Surface 1 along the X direction increases with increasing expansion gap depth. Furthermore, in Cases 1, 8, and 9, featuring varying collecting block thicknesses, the maximum expansion of Surface 1 along the X direction increases with increasing collecting block thickness. Owing to the small size of the heat collector block, the overall thermal expansion along the X direction ranges from 3.2 × 10^−3^ to 7.2 × 10^−3^ mm.

Figure 13 illustrates the thermal expansion of Surface 1 in the X direction along Line 1–6. In particular, for Line 1, the thermal expansion along the X direction presents fluctuating values in the middle, ranging from 3.2 × 10^−3^ to 3.4 × 10^−3^ mm. In contrast, at the upper and lower edges of the heat-collecting block, the thermal expansion increases to 3.3 × 10^−3^ to 3.5 × 10^−3^ mm. Under the same thickness of the heat-collecting block, the thermal expansion of Surface 1 along the X direction gradually increases from Lines 1 to 3, as shown in Figure 13a.

For Lines 4 to 6 (Figure 13b), which run along the thickness direction of the heat-collecting block, thermal expansion increases with increasing proximity of the heat-collecting block to the flame. Accordingly, the thermal expansion along Line 6 at a specific position is greater than the thermal expansions along other lines at the same position. Furthermore, the relationship between the thickness of the heat-collecting block and thermal expansion appears approximately linear.

### 4.3. Heat Transfer Coefficient (h)

Figure 14 displays the distribution of the heat transfer coefficient on Surface 1. Notably, large heat transfer coefficients are observed at locations along the edges of Surface 1. Ideally, according to the heat transfer process between the flue gas and heat-collecting block, the heat transfer coefficient of the lower section and fireside of the heat-collecting block should generally exceed that on the upper section and backfire side. However, in Case 3, the heat transfer coefficient on the fireside is generally lower than that on the backfire side. This is mainly due to the increased thermal resistance caused by the expansion gap located on the fireside.

Among the cases examined, the maximum heat transfer coefficient of Case 2 is lower than that of Case 1, whereas the maximum heat transfer coefficient of Case 3 exceeds those of both Cases 1 and 2. Across Cases 2, 6, and 7, the maximum heat transfer coefficient decreases with increasing expansion gap depth. Meanwhile, across Cases 2, 4, and 5, the maximum heat transfer coefficient first increases and subsequently decreases with increasing expansion gap width. Similarly, across Cases 1, 8, and 9, the maximum heat transfer coefficient first increases and then decreases with increasing thickness of the heat-collecting block.

Figure 15 indicates that along the vertical direction of Surface 1, the heat transfer coefficient decreases progressively from Lines 1 to 3, resembling the trend of thermal expansion along the X direction. In particular, near the upper and lower edges of the heat-collecting block, heat transfer coefficients are relatively high, reaching up to 1200 W/(m^2^·K). Meanwhile, under different working conditions, the heat transfer coefficients in the middle section of Line 2 remain almost constant, fluctuating between 490 W/(m^2^·K) and 520 W/(m^2^·K). However, in Cases 8 and 9, the heat transfer coefficients along Line 2 are the lowest. Meanwhile, the heat transfer coefficients along Lines 1 and 3 range from 569 W/(m^2^·K) to 1390 W/(m^2^·K) and from 390 W/(m^2^·K) to 841 W/(m^2^·K), respectively. In Cases 8 and 9, the heat transfer coefficients along Lines 1 and 3 are intermediate. In Cases 3 and 6, the heat transfer coefficients along Line 1 are the largest and smallest, whereas those along Line 3 are the smallest and largest, respectively.

Along Lines 4–6, heat transfer coefficients are higher on either side of the heat-collecting block and lower in the middle, as shown in Figure 15b. The maximum heat transfer coefficients along Lines 4, 5, and 6 are approximately 2200 W/(m^2^·K), 1500 W/(m^2^·K)), and 2400 W/(m^2^·K), respectively. Overall, the heat transfer coefficient of the heat-collecting block is the largest near the water wall, and it initially decreases and then increases with the proximity of the heat-collecting block to the fireside. In Cases 8 and 9, the heat transfer coefficients along Lines 4–6 are the lowest, followed by those in Case 3. Particularly in Cases 8 and 9, the heat transfer coefficients along Lines 4 and 6 decrease gradually after hitting their lowest limits.

### 4.4. Thermal Stress (σ)

Thermal stress distributions on Surface 1 are illustrated in Figure 16. Similar to the surface heat transfer coefficient, the thermal stress distribution on Surface 1 also exhibits higher values around the surface with lower values at its center. Notably, the maximum thermal stress is primarily concentrated at the four corners of Surface 1. Furthermore, the maximum thermal stress in Case 2 is lower and higher near the upper and lower edges of the fireside, respectively, while that in Case 3 is higher and concentrated in the upper and lower corners of the back fireside. Across Cases 2, 6, and 7, featuring varying expansion gap depths, the maximum thermal stress first decreases and then increases with increasing depth of the expansion gap, gradually approaching the fireside. Across Cases 2, 4, and 5, featuring varying expansion gap widths, the maximum thermal stress decreases with increasing width of the expansion gap. Across Cases 1, 8, and 9, featuring varying heat-collecting block thicknesses, the maximum thermal stress decreases with increasing thickness of the heat-collecting block.

Figure 17 indicates that the overall thermal stress along Line 1 is higher across different cases, with Case 3 exhibiting larger thermal stresses along both Lines 1 and 2. In particular, the thermal stress along Lines 1–3 ranges from 1.02 × 10^6^ Pa to 6.57 × 10^8^ Pa. Meanwhile, along Lines 4–6, the thermal stress ranges from 9.05 × 10^5^ to 5.75 × 10^7^ Pa, resembling the “U-shaped” distribution trend demonstrated by the heat transfer coefficient. Overall, the thermal stresses in Cases 8 and 9 are relatively small. Across the nine cases under investigation, the highest temperature of Surface 1 reached 430 °C. Based on the interpolation method, the yield strength and allowable stress for 15CrMo were obtained as 211 MPa and 118 MPa, respectively, both exceeding the simulated thermal stress of Surface 1.

### 4.5. Parameter Comparison and Structural Selection

Figure 18 displays the magnitude of thermal expansion along the X direction, heat transfer coefficient, thermal stress (obtained after taking the weighted average of the area of Surface 1), and temperature (obtained after taking the weighted average of the area at the temperature sensor measuring end). Notably, the variation trend of the thermal expansion along the X direction (black curve) contrasts that of the thermal stress (green curve). Meanwhile, the variation trend of thermal stress (green curve) resembles that of the heat transfer coefficient (red curve). Initially, the heat-collecting block and water-wall tubes are closely fitted. Upon exposure to high-temperature flue gas, these components experience varying degrees of expansion owing to temperature differences. Notably, thermal stress is positively correlated with the binding force between these components. For instance, a smaller expansion amount results in greater constraint, which, in turn, leads to greater thermal stress. This higher thermal stress indicates a greater degree of fit between the heat-collecting block and water-wall tubes, resulting in a larger heat transfer coefficient.

Notably, the temperature at the center of the water-wall tube on the fireside is 410 °C. Meanwhile, across all the nine cases under investigation, the temperatures at the measuring end of the heat-collecting block exceed 430 °C. Ideally, selecting the structure with the lowest simulated temperature at the measuring end of the heat-collecting block would yield the most accurate measurement results in practical scenarios. However, owing to variations in the processing accuracy of the heat-collecting block and wear conditions of the water-wall tube, the actual thermal resistance between the heat-collecting block and water-wall tube will likely exceed that predicted by the numerical simulation. Consequently, the temperature measured at the end of the heat-collecting block will exceed that derived from the numerical simulation. Hence, a structure with a lower temperature at the measuring end of the heat-collecting block based on the simulation results must be selected to ensure measurement accuracy and reduce the temperature of the temperature sensor, thus extending its service life. In Cases 1, 7, 8, and 9, the lower temperature observed at the measuring ends is attributed to the presence of small or no expansion gaps, which reduce the thermal resistance between the water wall and heat-collecting block. In practical installation scenarios, the degree of fitting between the heat-collecting block and water-wall tubes may not be ideal. Hence, to accommodate and address any potential variations in their arrangement, including an expansion gap is essential. This expansion gap can be compressed when necessary to ensure a close fit between the components and thus enhance heat transfer. Although Cases 1, 8, and 9 present a low temperature at the measuring end of the heat-collecting block, their lack of expansion gap makes them unsuitable for practical application. Conversely, Cases 3 and 6 present a higher temperature at the measuring end of the heat-collecting block. Furthermore, Case 7 presents a narrow expansion gap that is difficult to compress externally. Among Cases 2, 4, and 5, Case 2 presents the lowest measuring end temperature, which is approximately 30 °C higher than that at the center of the water-wall tube, and is thus selected as the ideal practical engineering application structure.

### 4.6. Verification and Application

For validation, the structure corresponding to Case 2 was applied to the water wall of a 1000 MW boiler in China. Five of the designed temperature-measurement devices (numbered 1~5) were installed on the inner surface of the furnace water-wall, as indicated in Figure 19. They were positioned at an elevation of approximately 55,268 mm, arranged in a counterclockwise order beginning from the front wall. To ensure the safety of the water-wall tubes, heat-collecting blocks and fixing bases were spot-welded with the fins. The overall field installation is depicted in Figure 2. Temperature sensors are in the process of rapid development, and many advanced sensors (such as flexible temperature sensors [27]) have been developed. However, considering practicality, installation convenience, post-maintenance, usage environment, cost, and durability, K-type thermocouples were used in current practical applications. The temperature measurement range (~1000 °C), accuracy (±0.4%), and response time (<1 s) of K-type thermocouples can meet the requirements of conventional industrial use. All data recorded by the device were transmitted to the distributed control system for display through an intelligent front end.

Figure 20 compares the average temperature readings obtained from the in-furnace water-wall temperature-measurement devices with those obtained from conventional temperature-measurement devices installed outside the furnace (out-furnace devices) under high, medium, and low load conditions. Notably, the temperatures recorded by the in-furnace temperature-measurement devices exceed those recorded by the out-furnace devices owing to flame radiation. Under high-load conditions, the maximum temperature recorded by the in-furnace temperature-measurement devices is approximately 505 °C, while that recorded by the out-furnace devices is approximately 409 °C, resulting in a temperature difference of less than 102 °C between the two. These results are consistent with those of previous studies [28], confirming the accuracy of the in-furnace temperature-measurement devices designed and developed in this study.

Figure 21 illustrates the temporal variation in the temperatures recorded by the in-furnace and out-furnace temperature-measurement devices under varying power-generation loads. Evidently, under an almost stable unit load, the fluctuations in the temperature data recorded by the out-furnace temperature-measurement devices are minimal, whereas those for the in-furnace temperature-measurement devices are more pronounced owing to greater effects of flame fluctuation and radiation heat transfer. Furthermore, under the same load, the differences between the temperature readings of different out-furnace temperature-measurement devices remain small. However, owing to combustion non-uniformity within the boiler, the differences between the temperature readings of different in-furnace temperature-measurement devices are significant, leading to some inaccuracy in the out-furnace devices. Notably, under high loads, in-furnace temperature-measurement devices 3 and 5 exhibit marked fluctuations in their temperature readings, with changes reaching approximately 150 °C. In contrast, the temperature data recorded by the out-furnace temperature-measurement devices remain relatively constant during this period. According to Li et al. [22], the observed temperature variation likely originates from alternating slagging and deslagging processes on the water wall. Furthermore, as illustrated in Figure 22, when the unit load decreases from 1000 MW to 700 MW and from 700 MW to 529 MW, the temperature variation range of the in-furnace devices becomes 3–92 °C, while that of the out-furnace devices remains at 7–36 °C. These findings underscore various advantages of the in-furnace water-wall temperature-measurement devices over their out-furnace counterparts: (a) greater sensitivity, (b) more accurate and intuitive reflection of the actual wall temperature inside the furnace, and (c) better indication of water-wall slag formation near the temperature-measurement devices. Remarkably, the developed in-furnace water-wall temperature-measurement devices have been operating normally for nearly 3 years.

## 5. Conclusions

In this study, an in-furnace water-wall temperature-measurement device was designed, and CFD methods were employed to simulate the thermal stresses and temperatures of heat-collecting blocks with different structures. Based on the simulation results, the thermal expansion, heat transfer coefficient, thermal stress, and temperature distribution characteristics of the contact surface between the heat-collecting blocks and water-wall tubes were analyzed. The optimal in-furnace water-wall temperature-measurement device was applied to a 1000 MW unit in China, providing a reference for direct measurements of inner water wall temperatures. Based on the numerical simulation and practical application analysis, the following conclusions were drawn:The temperature-measurement performance of various in-furnace water-wall temperature-measurement devices with different structures was examined by varying the thickness of the heat-collecting block and the location, width, and depth of the expansion gap. Although the temperatures recorded at the measuring end of the temperature sensor in Cases 1, 7, 8, and 9 were low, they were closest to the temperature of the water-wall tube. To ensure a tight fit between the water wall and heat-collecting block during actual installation and improve measurement accuracy while preventing damage to the water-wall tubes, the structure corresponding to Case 2 was selected as the optimal one. Notably, this structure features an expansion gap with a width of 0.5 mm and depth of 4 mm, and the measured end temperature is 30 °C higher than that at the center of the water wall.This optimal structure was applied to a 1000 MW unit in China to monitor the inner water-wall temperature. Notably, the measured data were more reliable compared to those recorded by conventional out-furnace water-wall temperature-measurement devices. Furthermore, the designed in-furnace water-wall temperature-measurement devices exhibited enhanced sensitivity, offering a more accurate representation of the actual temperature distribution across different regions of the water wall. These devices also offered insights into slag formation on the water wall and exhibited an extended service life. These findings offer valuable guidance for the practical operations of boilers.

## Figures and Tables

**Figure 1 sensors-24-06038-f001:**
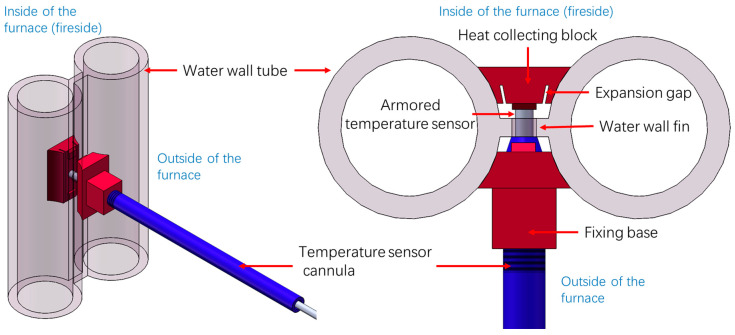
Temperature measurement device for the inner water wall of a boiler.

**Figure 2 sensors-24-06038-f002:**
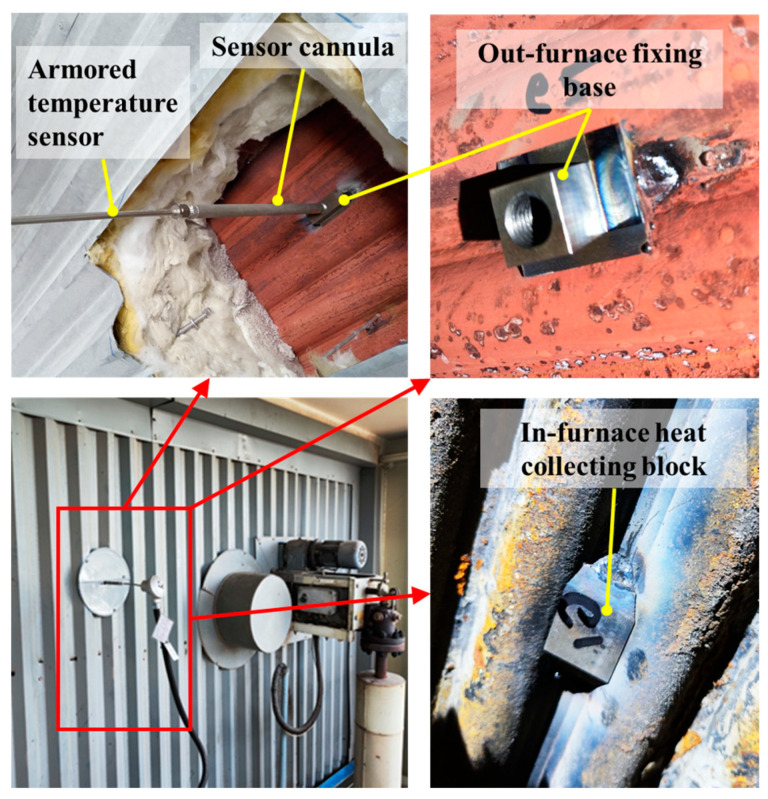
Application site installation diagram.

**Figure 3 sensors-24-06038-f003:**
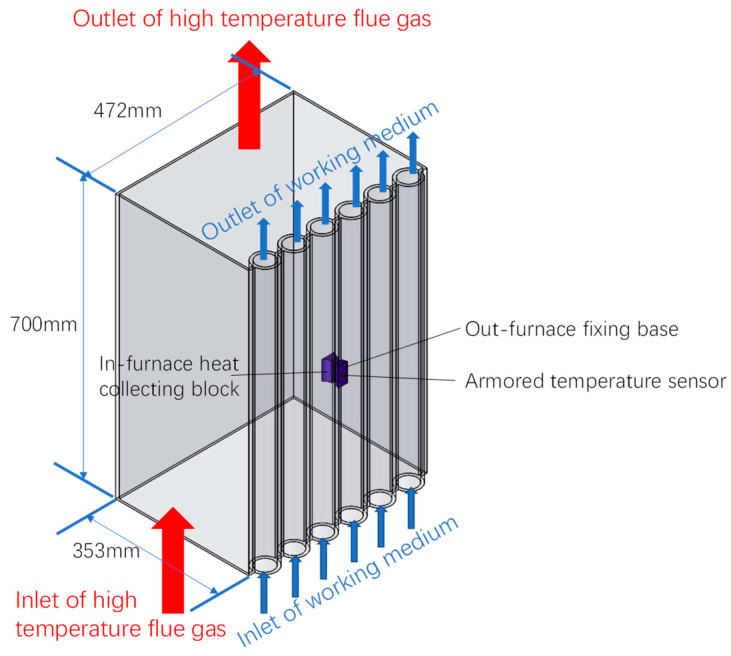
Numerical simulation domain.

**Figure 4 sensors-24-06038-f004:**
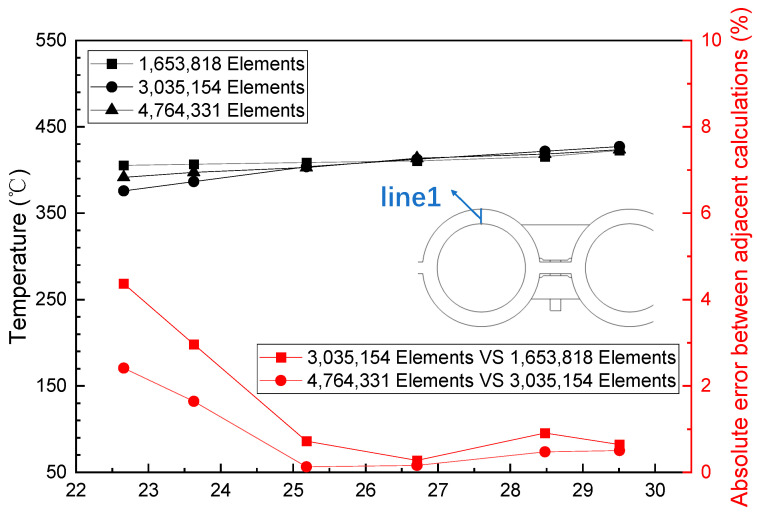
Grid independence verification of the water-wall tubes.

**Figure 5 sensors-24-06038-f005:**
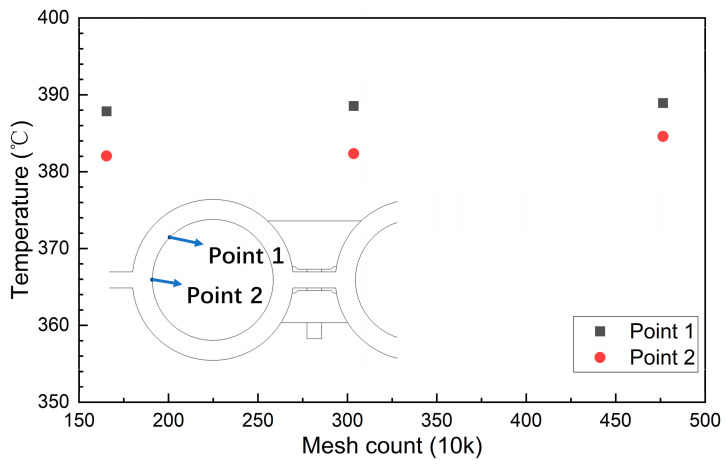
Mesh independence verification of the water wall.

**Figure 6 sensors-24-06038-f006:**
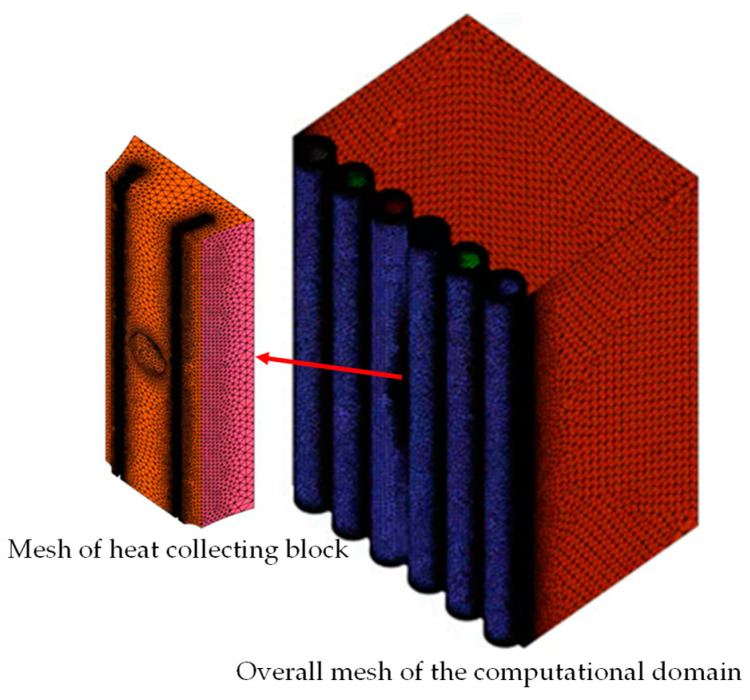
Mesh of the computational domain.

**Figure 7 sensors-24-06038-f007:**
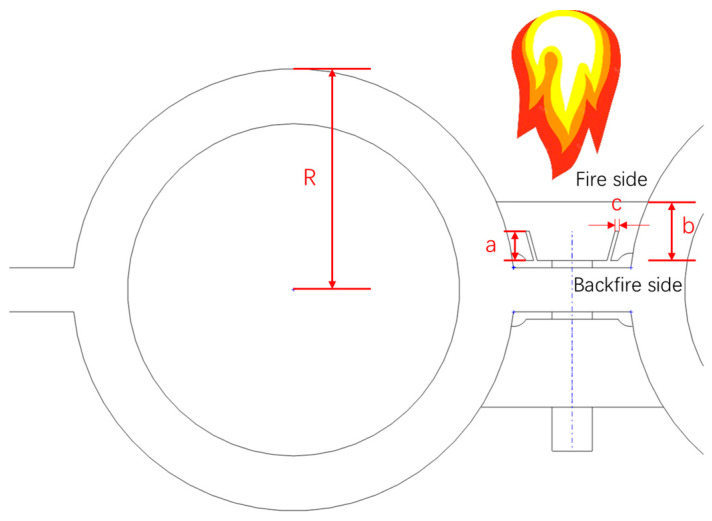
Specification of the heat-collecting block size.

**Figure 8 sensors-24-06038-f008:**
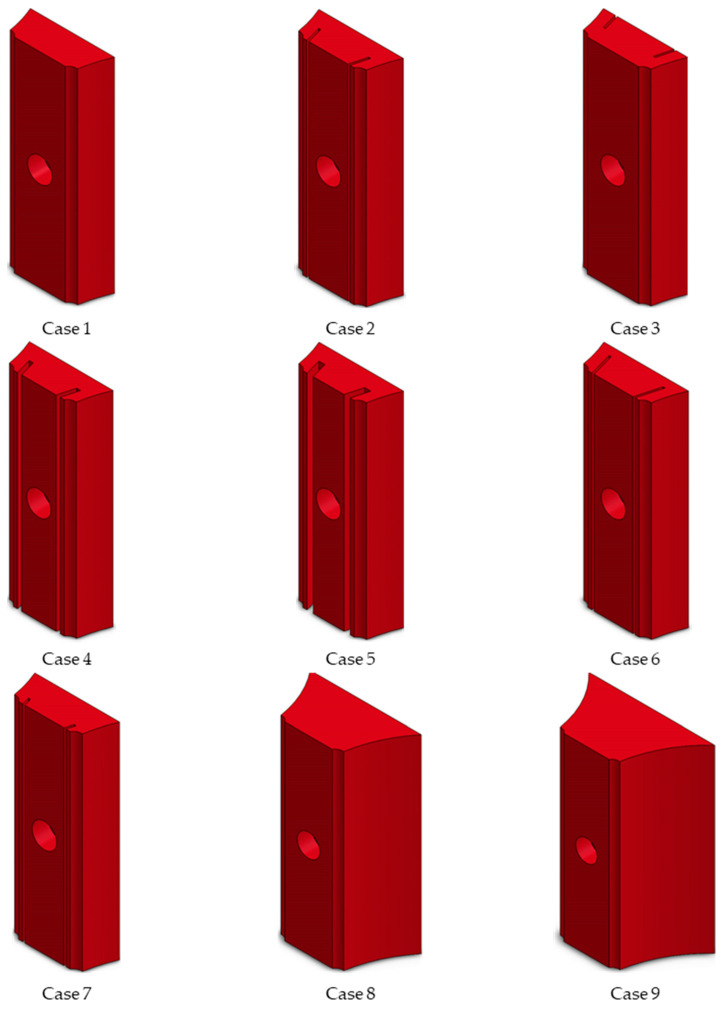
Structure of the heat-collecting block under different working conditions.

**Figure 9 sensors-24-06038-f009:**
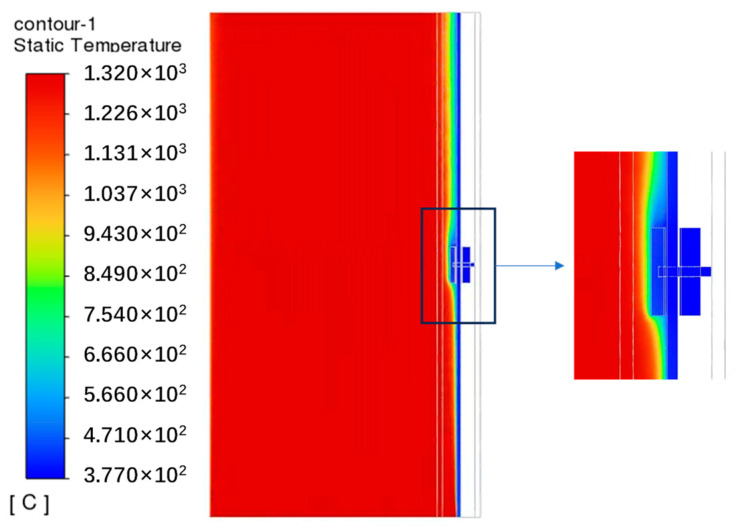
Flue gas temperature distribution at the vertical interface of the domain.

**Figure 10 sensors-24-06038-f010:**
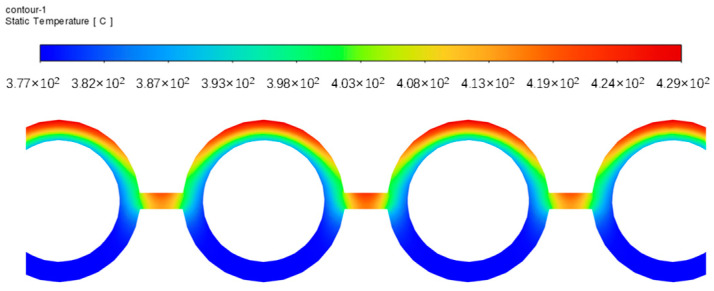
Temperature distribution at the top of the water wall.

**Figure 11 sensors-24-06038-f011:**
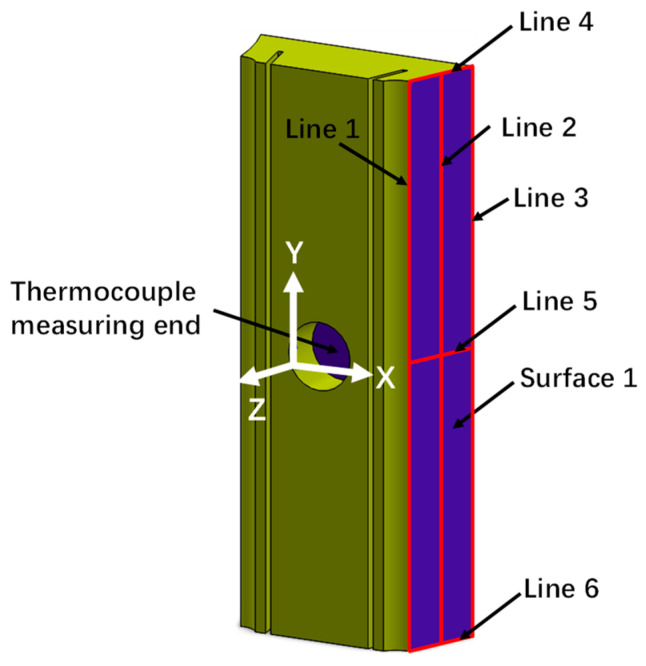
Key positions along the heat-collecting block.

**Figure 12 sensors-24-06038-f012:**
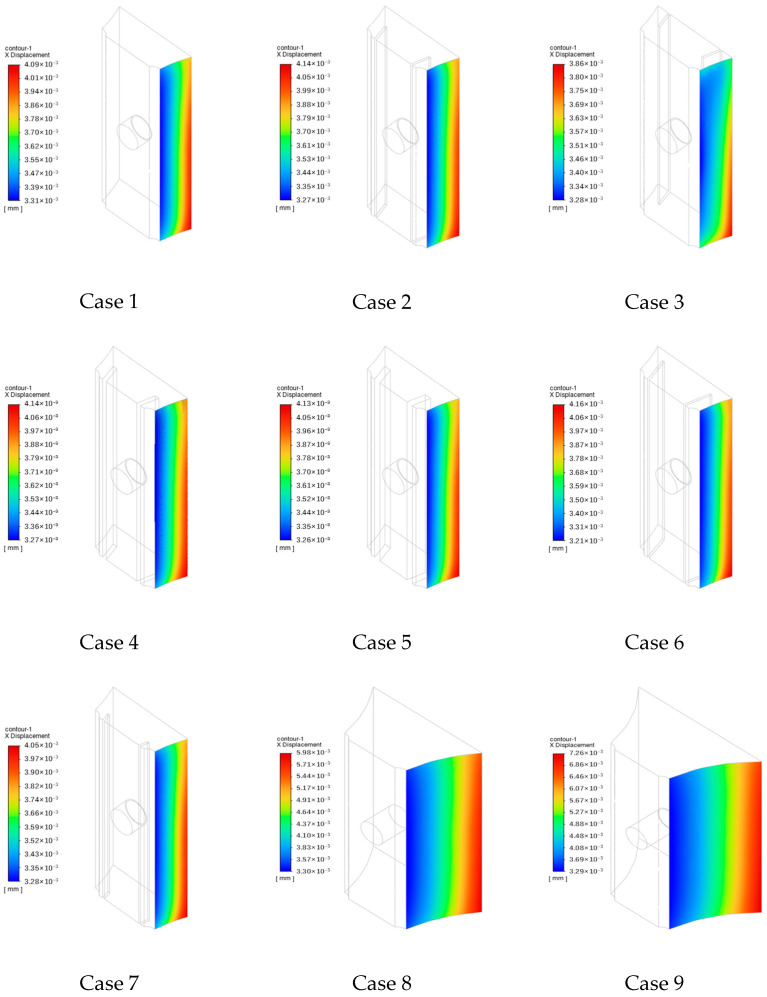
Thermal expansion of Surface 1 along the X direction.

**Figure 13 sensors-24-06038-f013:**
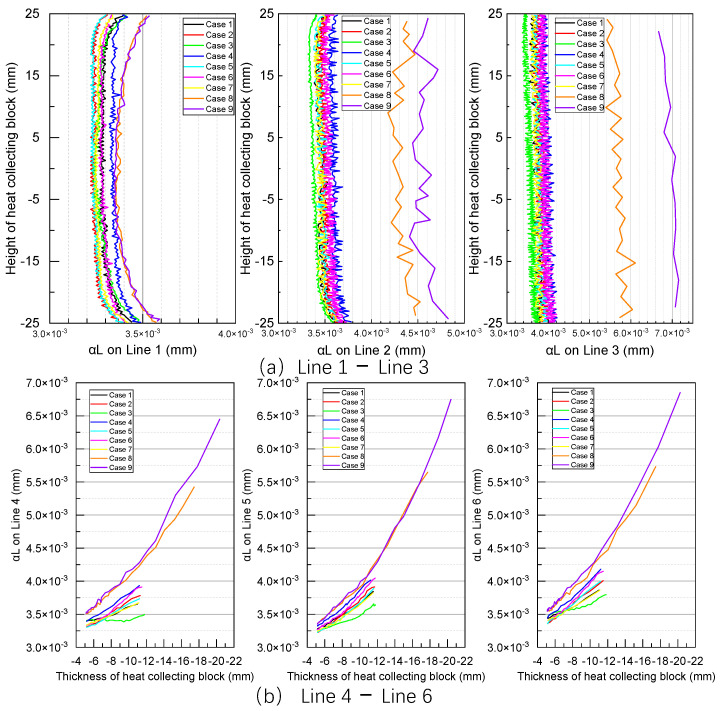
Calculation results of thermal expansion on Line 1–6.

**Figure 14 sensors-24-06038-f014:**
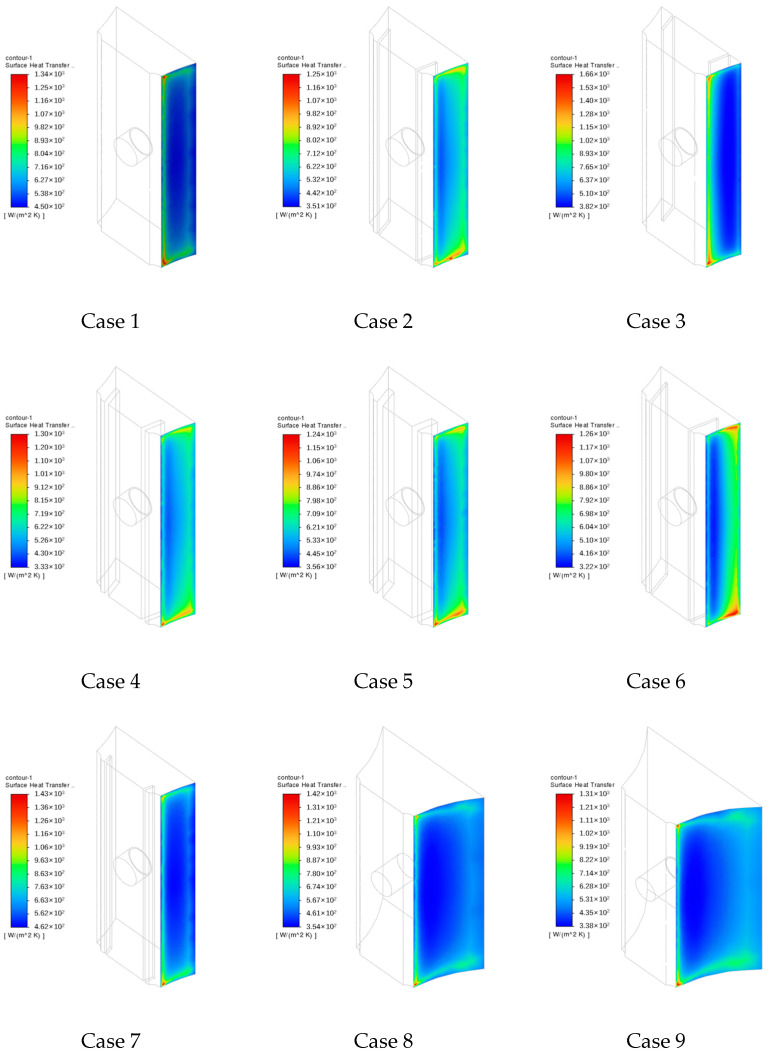
Heat transfer coefficients on Surface 1.

**Figure 15 sensors-24-06038-f015:**
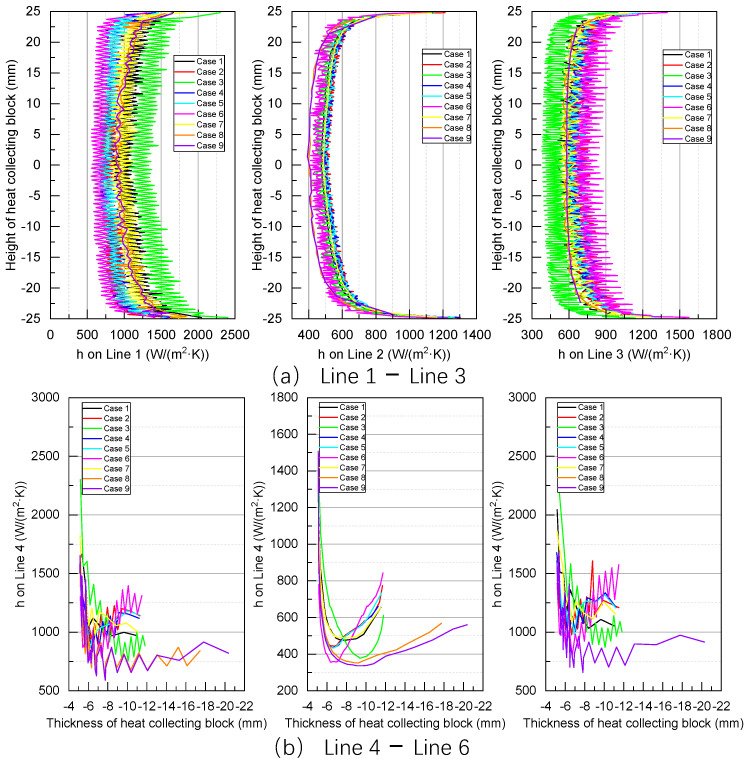
Calculation results of heat transfer coefficients on Lines 1–6.

**Figure 16 sensors-24-06038-f016:**
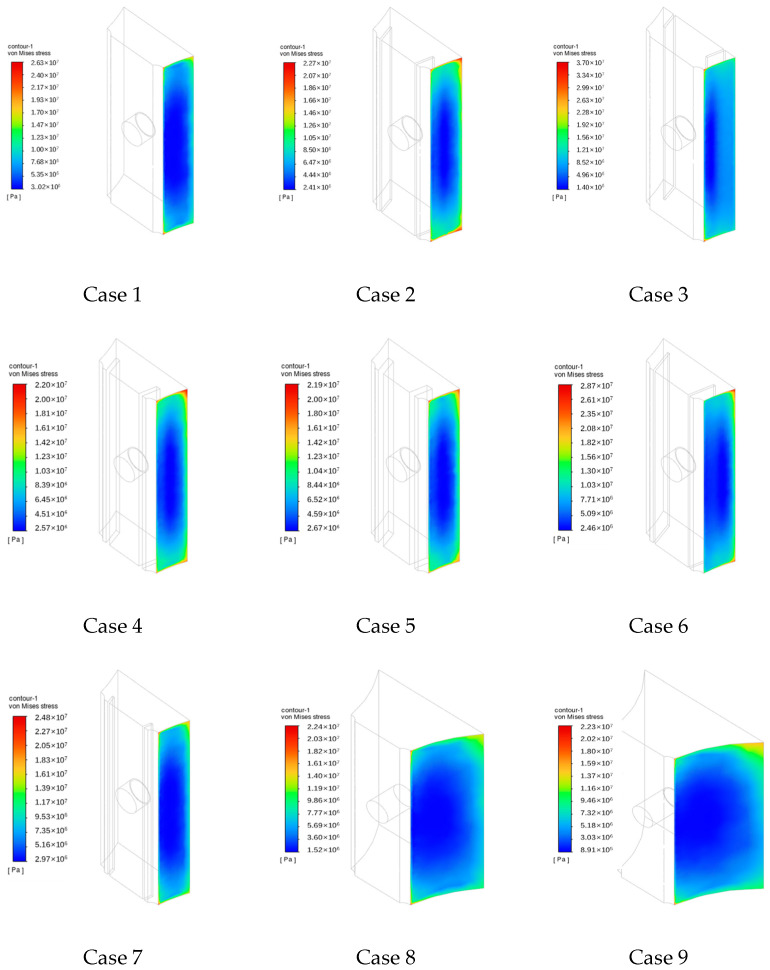
Thermal stress distributions on Surface 1.

**Figure 17 sensors-24-06038-f017:**
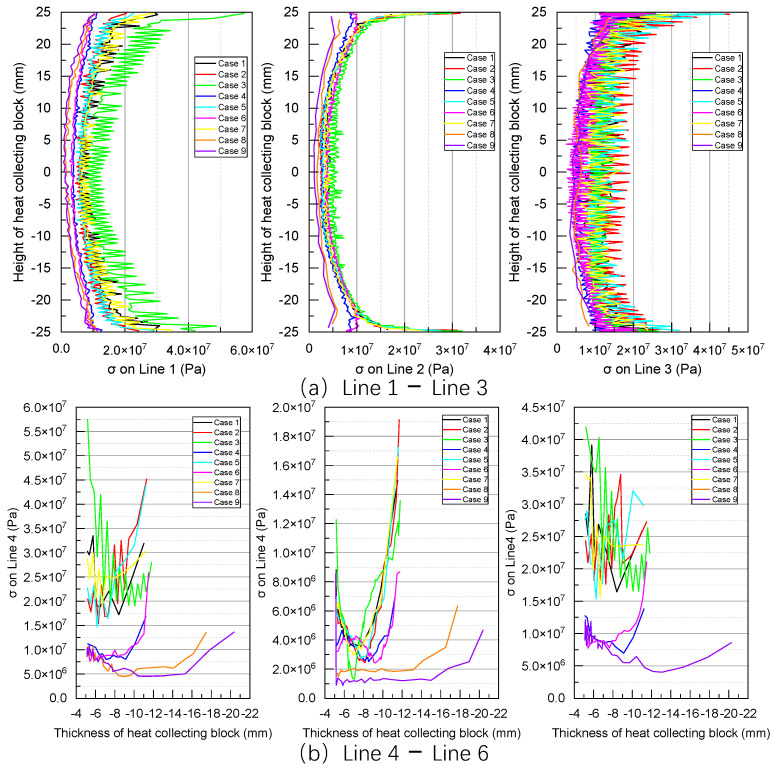
Calculation results of von mises stress distributions on Lines 1–6.

**Figure 18 sensors-24-06038-f018:**
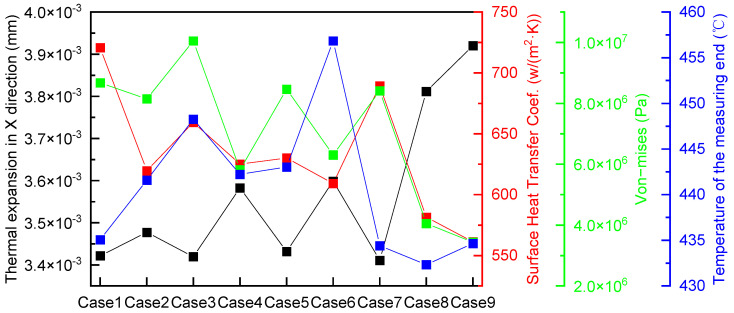
Comparison of key parameters.

**Figure 19 sensors-24-06038-f019:**
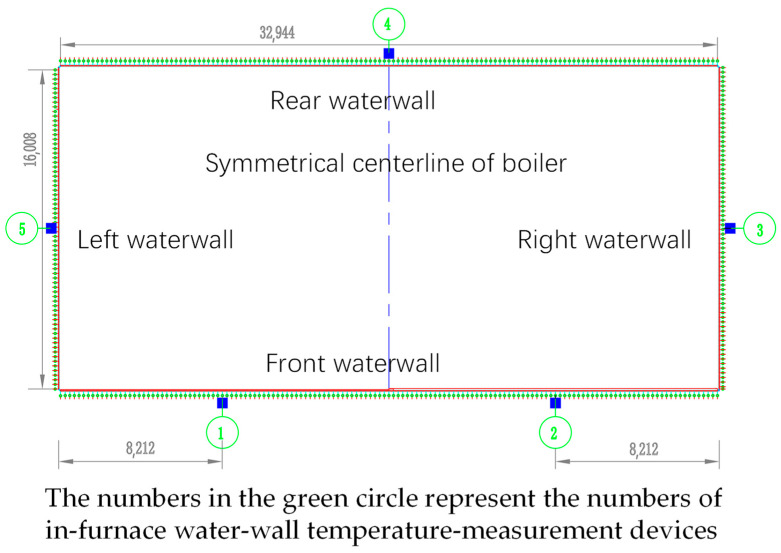
Locations of temperature-measurement devices installed on the inner surface water-wall.

**Figure 20 sensors-24-06038-f020:**
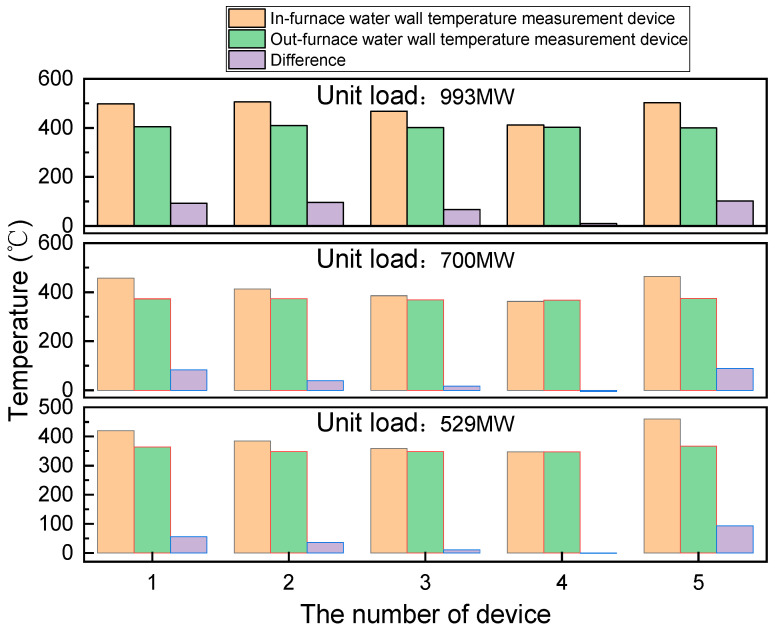
Comparison between temperature readings of the in-furnace and out-furnace water-wall temperature-measurement devices.

**Figure 21 sensors-24-06038-f021:**
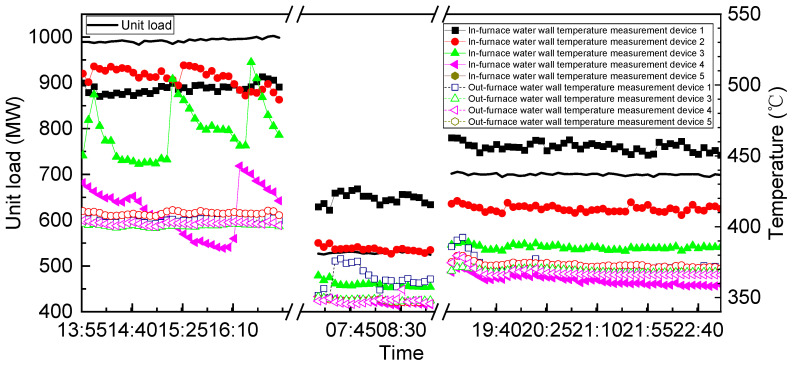
Temporal variation in the temperature readings of the in-furnace and out-furnace temperature measurement devices.

**Figure 22 sensors-24-06038-f022:**
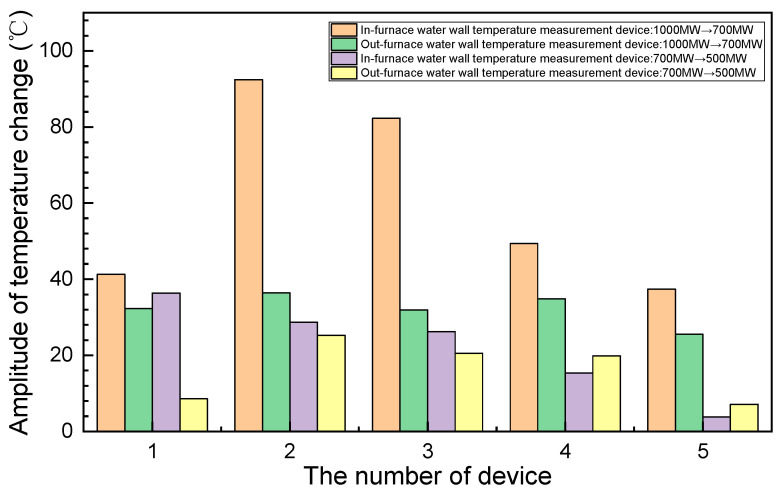
Amplitude of change in temperature with unit load.

**Table 1 sensors-24-06038-t001:** Key dimensions of the membrane water walls of the model established in this study.

Name	Value
Inner diameter of a single tube (mm)	45.3
Tube wall thickness (mm)	7.5
Tube pitch (mm)	76
Fin width (mm)	16
Fin thickness (mm)	6

**Table 2 sensors-24-06038-t002:** Chemical composition of 15CrMo steel (mass percentage, %) [23].

C	Si	Mn	P	S	Cu	Ni	Mo	Cr	Fe
0.15	0.27	0.55	0.03	0.03	0.03	0.03	0.50	0.95	Bal.

**Table 3 sensors-24-06038-t003:** Static parameters of 15CrMo steel [24].

	20 °C	100 °C	200 °C	300 °C	400 °C	500 °C	600 °C
Yield Strength (MPa)	295	-	269	242	216	198	-
Allowable Stress (MPa)	147	-	-	143	128	96	-
Elastic modulus (GPa)	206	199	190	181	172	163	-
Thermal expansion coefficient (10^−6^/K)	-	11.9	12.6	13.2	13.7	14.0	-
Thermal conductivity (W/m·K)	-	40.6	40.1	38.7	36.8	34.8	32.8
Density (kg/m^3^)	7800	-	-	-	-	-	-
Specific heat capacity (J/(kg·K)	-	-	590	607	657	712	800
Poisson’s ratio	0.284	0.295	0.300	0.301	0.304	0.308	-

**Table 4 sensors-24-06038-t004:** Conditions adopted for the numerical simulation.

Condition Number	Thickness of Heat Collecting Block (b, mm)	Expansion Gap		
		Location	Width (c, mm)	Depth (a, mm)
Case 1	26%R	/	/	/
Case 2	26%R	Backfire side	0.5	4.0
Case 3	26%R	Fireside	0.5	4.0
Case 4	26%R	Backfire side	1.0	4.0
Case 5	26%R	Backfire side	1.5	4.0
Case 6	26%R	Backfire side	0.5	5.8
Case 7	26%R	Backfire side	0.5	1.9
Case 8	50%R	/	/	/
Case 9	60%R	/	/	/

## Data Availability

Dataset available on request from the authors.

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
