# Peer review of "Numerical Study and Structural Optimization of Water-Wall Temperature-Measurement Device for Ultra-Supercritical Boiler"

_sensors, 2024, doi:10.3390/s24186038_

Round 1

Reviewer 1 Report

Comments and Suggestions for Authors

This article proposes a method for measuring the temperature inside a water-cooled furnace using temperature measurement points arranged inside a supercritical boiler water-cooled furnace. By installing temperature measurement devices inside the water-cooled walls of the furnace, several different structural forms were compared for measuring the temperature of the water-cooled walls inside the furnace. This result has certain reference value for the measurement arrangement of water wall temperature in the furnace of large-scale coal-fired units in the future. The practical work of this paper indeed reflects the current demand and exploration work for measuring the temperature of water-cooled walls during the operation of coal-fired boilers.

The measurement method used is indeed different from the current coal-fired boiler water-cooled wall, which uses the temperature measurement point on the back side of the water-cooled wall to obtain the actual operating situation of the water-cooled wall inside the furnace. This is because the arrangement of measurement points outside and inside the furnace is different, and the temperature of the water-cooled wall inside the furnace can be obtained based on the changes outside the furnace to obtain relevant information. The exploration of the paper reflects the author's research exploration in this area, with the goal of obtaining the temperature of the water-cooled wall directly inside the furnace. The temperature of the water-cooled wall in the furnace obtained in the paper mainly reflects the research results of this paper, and has not been compared with other measurement or experimental research results, which provides limited support for the method used in the paper. If wall temperature measurement points are installed on site in this paper, it is recommended that the author add temperature measurement points inside the furnace to compare the results with the measurement results proposed in this paper.

Although in practical work, the measuring points arranged inside the furnace have limited operating time due to the furnace environment, the initial results can indeed help support the measurement method proposed in this paper. It is suggested that the author compare these temporary temperature measurement points with the results of this article, which can effectively support the feasibility and reliability of the measurement method proposed in this article. The measurement method proposed in this paper essentially reflects the temperature of the metal on the water-cooled wall of the boiler. However, how these measurements can truly reflect the metal temperature on the water-cooled wall of the boiler cannot fully support these conclusions. It is suggested that the author add temporary wall temperature measurement points inside the water-cooled wall and compare the accuracy of the measurement methods and data proposed in this method, which will help the paper gain universal recognition and approval.

Author Response

Comment :This article proposes a method for measuring the temperature inside a water-cooled furnace using temperature measurement points arranged inside a supercritical boiler water-cooled furnace. By installing temperature measurement devices inside the water-cooled walls of the furnace, several different structural forms were compared for measuring the temperature of the water-cooled walls inside the furnace. This result has certain reference value for the measurement arrangement of water wall temperature in the furnace of large-scale coal-fired units in the future. The practical work of this paper indeed reflects the current demand and exploration work for measuring the temperature of water-cooled walls during the operation of coal-fired boilers.

The measurement method used is indeed different from the current coal-fired boiler water-cooled wall, which uses the temperature measurement point on the back side of the water-cooled wall to obtain the actual operating situation of the water-cooled wall inside the furnace. This is because the arrangement of measurement points outside and inside the furnace is different, and the temperature of the water-cooled wall inside the furnace can be obtained based on the changes outside the furnace to obtain relevant information. The exploration of the paper reflects the author's research exploration in this area, with the goal of obtaining the temperature of the water-cooled wall directly inside the furnace. The temperature of the water-cooled wall in the furnace obtained in the paper mainly reflects the research results of this paper, and has not been compared with other measurement or experimental research results, which provides limited support for the method used in the paper. If wall temperature measurement points are installed on site in this paper, it is recommended that the author add temperature measurement points inside the furnace to compare the results with the measurement results proposed in this paper.

Although in practical work, the measuring points arranged inside the furnace have limited operating time due to the furnace environment, the initial results can indeed help support the measurement method proposed in this paper. It is suggested that the author compare these temporary temperature measurement points with the results of this article, which can effectively support the feasibility and reliability of the measurement method proposed in this article. The measurement method proposed in this paper essentially reflects the temperature of the metal on the water-cooled wall of the boiler. However, how these measurements can truly reflect the metal temperature on the water-cooled wall of the boiler cannot fully support these conclusions. It is suggested that the author add temporary wall temperature measurement points inside the water-cooled wall and compare the accuracy of the measurement methods and data proposed in this method, which will help the paper gain universal recognition and approval.

Response: Thank you very much for your comments and professional advice. As you have suggested, if some temporary temperature measurement points were added at corresponding positions inside the furnace, some useful temperature data in limited operating time could be gotten to effectively support the feasibility and reliability of the measurement method proposed in this article and further improve the academic rigor of our article.

However, installing the temporary temperature measurement points at corresponding positions requires the coal-fired unit to be shut down and a large lifting platform to be built inside the furnace. Since the coal-fired unit is now in regular operation, it is very difficult to add the temporary temperature measurement points in the short term. If there is any opportunity in future projects, we would install temporary temperature measurement points according to your suggestion to further support the research in this article.

In Section 4.6 of this paper, we compared the average temperature readings obtained from our in-furnace water-wall-temperature-measurement device with those obtained from conventional temperature-measurement devices installed outside the furnace. In this way, the accuracy and advantages of our in-furnace temperature-measurement device over conventional temperature-measurement devices are confirmed to a certain extent.

Reviewer 2 Report

Comments and Suggestions for Authors

A temperature measuring device for boiler temperature detection is designed and tested in practical environment. It has strong engineering significance. However, there are still many problems with the manuscript, and the current version cannot be accepted.

1. In the introduction, the author lists previous works, but lacks intuitive data comparison to highlight the advantages of this manuscript, and suggests adding comparison tables or images. In addition, there are other papers that can be referred to for the work of temperature sensors, such as: International Journal of Extreme Manufacturing, 2023, 5(1): 015601. 

2. The author uses too many pictures in the text, even reaching Figure 28. Consider putting some images into the supplementary information section/or stitching images together. (e.g. grid divided images, etc.)

3. The author should add a physical diagram of the actual installation of the sensor (move Figure 25 forward). At the same time, the actual installation structure of the module should be shown, and the detailed picture of the structure should be provided. According to the author's test results, the temperature was measured both inside and outside the furnace, but I did not see the installation position of the temperature sensor outside the furnace in Figure 1.

4. The annotation in Figure 2 is incorrect.

4. In the simulation results shown in Figure 8-12, are the relevant boundary conditions for water flow in the pipe added?

5. It is recommended to change the color of Figure 13. The color of the axes in the figure is not easy to distinguish.

6. The author has systematically simulated the thermal stress of the structure (by changing the structure), but has it been verified by strain gauges in the actual test?

7. What type of armored thermocouple does the author use?

Comments on the Quality of English Language

Language needs to be optimized.

Author Response

Dear  Reviewer:

Thank you for your letter and comments concerning our manuscript entitled “Numerical study and structural optimization of water wall temperature measurement device for ultra-supercritical boiler” (Manuscript ID: sensors-3185844). The insightful comments and suggestions have been very helpful in improving our paper. We have studied the reviewers’ comments carefully and made suggested corrections. The main changes made as well as our responses to the reviewers’ comments are itemized as follows:

Reviewer #2:

A temperature measuring device for boiler temperature detection is designed and tested in practical environment. It has strong engineering significance. However, there are still many problems with the manuscript, and the current version cannot be accepted.

 [Comment 1]: In the introduction, the author lists previous works, but lacks intuitive data comparison to highlight the advantages of this manuscript, and suggests adding comparison tables or images. In addition, there are other papers that can be referred to for the work of temperature sensors, such as: International Journal of Extreme Manufacturing, 2023, 5(1): 015601.

[Response]: Thank you very much for your suggestion. The introduction has been partially revised according to your suggestions. However, the authors has consulted numerous literatures and found that there is relatively little research on direct measurement of the temperature of water-wall in the furnace, making it difficult to summarize temperature data or charts that can be used for direct comparison. Compared with other measurement methods, the direct measurement device for the inner wall temperature of water-wall designed in this article is an innovative approach. In practical applications, selecting K-type temperature sensors as measuring elements can meet industrial needs. This method has the advantages of low cost, easy installation, long service life, accurate measurement, and the ability to be widely deployed.

In addition, comparing the data with the original outer wall temperature measurement points of the boiler in Section 4.6 can also highlight the advantages of this study.

Thank you again for your suggestions on the references, which have been cited in the article. Flexible temperature sensors are of great help in temperature measurement, and if there are suitable opportunities in engineering applications in the future, we will delve into understanding and even use them.

[Comment 2]: The author uses too many pictures in the text, even reaching Figure 28. Consider putting some images into the supplementary information section/or stitching images together. (e.g. grid divided images, etc.)

[Response]: Thank you very much for your suggestion. We have made adjustments to the pictures based on your suggestions. In the revised version of our manuscript, some pictures have been stitched together and some pictures have been moved into supplementary information.

[Comment 3]: The author should add a physical diagram of the actual installation of the sensor (move Figure 25 forward). At the same time, the actual installation structure of the module should be shown, and the detailed picture of the structure should be provided. According to the author's test results, the temperature was measured both inside and outside the furnace, but I did not see the installation position of the temperature sensor outside the furnace in Figure 1.

[Response]: Thank you very much for your suggestion. In the revised version of our manuscript, we have moved Figure 25 and related descriptions from Section 5 to Section 2.2. Besides, some modification has been made to the picture to better show the details of on-site installation.

Our device was designed to directly obtain the temperature of the water-wall inside the furnace, so there is no installation position of the temperature sensor outside the furnace in Figure 1. The out-furnace water-wall temperature data mentioned in Section 4 (Results and Discussion) were not obtained by our device, but were obtained using traditional measurement methods. And the relevant temperature sensors were installed outside the furnace during the construction of the boiler.

[Comment 4]: The annotation in Figure 2 is incorrect.

[Response]: Thank you so much for pointing out. In the revised version of our manuscript, the annotation in Figure 2 has been corrected.

[Comment 5]: In the simulation results shown in Figure 8-12, are the relevant boundary conditions for water flow in the pipe added?

[Response]: Thank you very much for your comment. During simulation, we added relevant boundary conditions for the working medium inside the pipe. And as described in Section 3.5, the boundary conditions of the working medium inside the pipe (temperature, flow rate, etc.) are defined based on the actual operating parameters of the selected 1000 MW ultra-supercritical boiler. Due to article space limitations, details of the defined boundary conditions are not listed.

[Comment 6]: It is recommended to change the color of Figure 13. The color of the axes in the figure is not easy to distinguish.

[Response]: Thank you so much for your suggestion. In the revised version of our manuscript, the color of the axes has been changed to white.

[Comment 7]: The author has systematically simulated the thermal stress of the structure (by changing the structure), but has it been verified by strain gauges in the actual test?

[Response]: Thank you so much for your comment. The thermal stress of the structure has not been verified by strain gauges in the actual test. Installing strain gauges at corresponding positions requires the unit to be shut down and a large lifting platform to be built inside the furnace. Since the coal-fired unit is now in regular operation, it is very difficult to add strain gauges in the short term.

The fluid-solid heat transfer and thermal stress simulation method used in this article is similar to that in reference [Zhang Z, Yang Z, Nie H, et al. A thermal stress analysis of fluid–structure interaction applied to boiler water-wall [J]. Asia-Pacific Journal of Chemical Engineering, 2020, 15(6): e2537], indicating that the simulation results in this article are relatively accurate. Besides, as of the reply date, we have installed our temperature measurement device for different boilers in multiple projects, with a maximum service life of over 3 years. This also demonstrates the feasibility of the the structure of our device. If there is any opportunity in future projects, we will add strain gauges according to your suggestion to further verify the thermal stress of the structure.

[Comment 8]: What type of armored thermocouple does the author use?

[Response]: Thank you very much for your comment. We used K-type thermocouples in practical applications, as described in Section 4.6 of the article.

The manuscript has been carefully reviewed by an experienced editor whose first language is English and who specializes in editing papers written by scientists whose native language is not English.

We look forward to hearing from you at your earliest convenience.

Sincerely,

Zifu Shi

Zhejiang University

No. 38 Zhejiang Road, Xihu District, Hangzhou City, Zhejiang Province Hangzhou, CN 310027

Phone No:+86-13073688785

Email Address: shizifu@zju.edu.cn

Round 2

Reviewer 2 Report

Comments and Suggestions for Authors

The author has revised the manuscript, and the current version is acceptable.